# Recent Advances in Reconfigurable Metasurfaces: Principle and Applications

**DOI:** 10.3390/nano13030534

**Published:** 2023-01-28

**Authors:** Ziyang Zhang, Hongyu Shi, Luyi Wang, Juan Chen, Xiaoming Chen, Jianjia Yi, Anxue Zhang, Haiwen Liu

**Affiliations:** 1Shaanxi Key Laboratory of Deep Space Exploration Intelligent Information Technology, Xi’an Jiaotong University, Xi’an 710049, China; 2School of Information and Communications Engineering, Xi’an Jiaotong University, Xi’an 710049, China

**Keywords:** reconfigurable metasurface, dynamic control, phase-change materials, liquid crystal, graphene, semiconductor components, MEMS

## Abstract

Metasurfaces have shown their great capability to manipulate electromagnetic waves. As a new concept, reconfigurable metasurfaces attract researchers’ attention. There are many kinds of reconfigurable components, devices and materials that can be loaded on metasurfaces. When cooperating with reconfigurable structures, dynamic control of the responses of metasurfaces are realized under external excitations, offering new opportunities to manipulate electromagnetic waves dynamically. This review introduces some common methods to design reconfigurable metasurfaces classified by the techniques they use, such as special materials, semiconductor components and mechanical devices. Specifically, this review provides a comparison among all the methods mentioned and discusses their pros and cons. Finally, based on the unsolved problems in the designs and applications, the challenges and possible developments in the future are discussed.

## 1. Introduction

Metamaterials are artificial materials with periodic or quasi-periodic, sub-wavelength-sized structures. Unlike natural materials, which have inherent electromagnetic properties, metamaterials can achieve novel electromagnetic properties, such as negative permittivity and permeability [1,2,3,4,5], by designing specific structured unit cells and arranging them in a periodic manner. Due to the bulky volume of metamaterials and the difficulty to fabricate them, a two-dimensional metamaterial, i.e., metasurface, was proposed on the basis of metamaterial [6]. Metasurfaces have the thickness of less than one wavelength and have the capability to manipulate the electromagnetic wave fronts to realize novel electromagnetic phenomenon.

Categorized by the two important properties of electromagnetic waves—amplitude and phase—metasurfaces can be divided into two types, amplitude modulation metasurfaces and phase modulation metasurfaces. Some metasurfaces that can manipulate both the parameters were reported [7,8,9,10]. Amplitude modulation metasurfaces could modulate the amplitude of electromagnetic waves in specific frequency bands. Some applications, such as perfect absorption, were reported [11,12,13,14,15,16].

Phase modulation metasurfaces manipulate the phase of electromagnetic waves while the amplitude response remains a constant. By configuring a specific phase gradient on the surface, reflections or transmissions in abnormal directions can be achieved. Based on phase modulation of the metasurface, beam focusing [17,18,19], scattering [20,21,22] and beam shaping [23,24,25] were realized.

Once the metasurfaces mentioned above are fabricated, the functions are fixed and, therefore, referred to as passive metasurfaces. With the continuous progress of metasurface research, some researchers are trying to realize novel dynamical functions. As a result, the concept of reconfigurability was introduced into the metasurfaces [26]. The reconfigurability is achieved by loading changeable structures that can change their states and properties under external excitations. When we apply different excitations, reconfigurable metasurfaces can achieve different electromagnetic responses. Due to engineering practice, considering the simplicity and costs of metasurfaces, researchers choose some response states from the continuous state range, and a range of response states are quantified as one state. For example, the phase response from 90 to 270 degrees is quantified as 180 degrees, and the phase response from 0 to 90 degrees and 270 to 360 degrees is quantified as 0 degrees, the quantification of the phase response of the 1-bit metasurface is realized. Similarly, when choosing more states, a 2-bit or multi-bits metasurface can be designed. Based on this principle, the concept of coding metasurfaces or digital metasurfaces is proposed. The coding metasurfaces build a bridge between the physical world and the digital world. Due to the quantification of states, the theoretical calculation of the far field of the coding metasurfaces is simplified, which makes the design process of the metasurfaces much easier.

Furthermore, when we change the states of metasurfaces in time domain, we can obtain different harmonic waves in the frequency domain, namely, the time-varying control of metasurfaces. Based on this principle, some studies have realized the frequency modulation of the incident wave, designed metasurface communication systems and successfully transmitted data [27,28,29,30,31,32,33,34,35,36,37,38,39,40,41,42,43,44].

Currently, there are many changeable structures for reconfigurable metasurfaces from the microwave band to the optical band. Due to the different properties of various changeable structures, we have different methods to control the changeable structures, including electrical, thermal, optical and mechanical ways. This review summarizes the development of reconfigurable metasurfaces enabled by different changeable structures or devices on reconfigurable metasurfaces and compares their advantages and disadvantages. By analyzing the shortcomings of various implementation methods, some feasible solutions are provided. Finally, the future development of the reconfigurable metasurface is conceived.

## 2. Novel Applications of Metasurfaces

As proposed above, the advent and development of metasurfaces present a new way to precisely manipulate electromagnetic (EM) waves. Now, researchers are engaging in designing multifunctional metasurfaces to adapt to more complicated application environments. All the applications demand qualified reconfigurable metasurfaces.

Metasurface are well-known for their manipulation of EM waves. Beam focusing and deflecting are one of the basic applications of the metasurface [45,46,47,48,49,50,51,52,53,54,55,56,57,58,59,60,61]. In the microwave band, there have been mature approaches using the phased arrays technique. Beam focusing can be achieved in any specified direction by applying step phase on antenna arrays. However, the microwave shifters, which implement the step phase, are usually expensive. In order to achieve high gains of the focused beams, large arrays are needed, resulting in high cost. In recent years, the development of terahertz wireless communication technology has put forward new requirements for the traditional communication technology. In the terahertz band, the wavelength is shorter and systems have a higher spatial resolution, thus more precise target recognition can be achieved. However, semiconductor devices have higher losses in the terahertz band and are, therefore, limited in applications. Using metasurfaces, beam focusing in any direction can be achieved by configuring phase gradients and is less expensive than traditional beam focusing techniques. Additionally, metasurfaces shows their great capability in polarization conversion [62,63,64,65,66,67,68,69,70,71,72]. Metasurfaces provide new degrees of freedom to realize complex EM functionalities.

Similar potential applications exist in current communication technologies. With the rapid development of 5G and the next-generation communication technologies, the working frequencies of communication systems are becoming higher and the sizes of antenna arrays are increasing, which also brings various challenges. In the next generation of wireless communication, how to realize high-speed communication is still under discussion. One idea is to use large-size antenna arrays to construct multiple input multiple output (MIMO) systems, requiring more radio frequency (RF) chains and antennas, which greatly increases the cost and occupies a large space. Another idea is to communicate in a higher frequency band, such as the terahertz band, but the path loss of signals rapidly deteriorates in higher frequency bands, which brings higher requirements for the designs of high-performance communication system hardware. The implementation of reconfigurable metasurfaces is expected to solve those challenges above. By applying specific control signals to the metasurfaces and carrying out time modulation on metasurfaces, the incident waves can be modulated in the frequency domain directly, which can replace modulators and transmitters in RF chains. The spatial modulation of metasurfaces can realize multiplexing in space diversity. Currently, there is some research that is based on the reconfigurable metasurfaces to establish communication systems, and researchers successfully realized data transmission between metasurfaces [30,41,73,74,75,76,77,78,79]. Metasurfaces are expected to innovate the framework of communication systems in the future.

In traditional optical imaging systems, such as cameras, light travels through complicated cascaded optical lenses to realize various functions. Those lenses in traditional optical imaging systems are burnished mechanically to focus light. Traditional optical lenses suffer from inherent spherical error, resulting in the distortion of the object image. Moreover, limited by the materials, traditional optical lenses are usually large and bulky. In order to change the focal point of an imaging system, we need to mechanically adjust the position of the lenses, so it is difficult to adjust the lenses quickly when accuracy is required due to the heavy weight of the lenses. In addition, the large volume of lenses constricted their usages in miniaturized equipment. Metasurface planar lenses eliminate the spherical error inherently, can achieve dynamic optical functions and reduce the volume of lenses [80,81,82,83,84,85,86,87,88,89].

Holography is another application of metasurface in optics. Holography was first used to record the object light in the form of diffraction stripes on specific mediums based on the principle of interference. The recorded diffraction stripes include information about the amplitude and phase of the object light. The original object light wave can then be reproduced by illustrating the medium by referencing light. There are inherent defects in this procedure, when the object light is difficult to obtain, optical holography cannot be recorded. Thus, the concept of computational holography appears. We can simulate diffraction patterns of object light in computer software based on computational holography principle, even when the object light does not actually exist in the physical world. Traditional holograms can only be reproduced through their record medias, they are fixed and passive, while reconfigurable metasurfaces can be used as adjustable recording media to record the information of amplitude and phase. By calculating the required amplitude and phase information on the metasurface, holograms can be generated under the exposure of the incident wave [47,90,91,92,93,94,95,96,97,98].

With the rapid development of the Internet of Things (IoTs), fast image processing techniques are urgently needed in those fields. Current digital image processing techniques rely on analog-to-digital converters to transform information from the physical world to the digital domain and then the sampled information is processed by integrated circuits or optical information systems. When processing huge amounts of images, small integrated circuits will lead to slow processing speed, but increasing the volume of integrated circuits will bring a lot of inconvenience to the portability of those IoT devices; The latter was discussed above, traditional optical imaging systems have many obstacles to be overcome for usage in current applications. Some researchers have designed metasurfaces to implement logical operation and optical information processing, which can directly manipulate the input signals in the analog domain to achieve the same function as the traditional optical information system [99,100,101,102,103].

In military applications, metasurfaces can achieve better performance than conventional technologies. Researchers have used metasurfaces to achieve absorbers, which have higher absorbing efficiency and wider operating bandwidth than conventional absorbing materials [104,105,106,107,108,109]. The metasurfaces reduce the radar scattering cross section of the antennas by deploying random phase responses on the surface so that the reflected waves disperse in all directions [62]. Due to the EM manipulation capability of metasurfaces, invisible cloaks for specific frequencies were realized [110,111]. Objects can be concealed beneath metasurfaces and regarded as flat surfaces under EM wave detection. Reconfigurable metasurfaces are expected to have a wider range of applications in the military field.

There are several ways to implement reconfigurable techniques for different applications and frequency regimes of metasurfaces. Here, we will introduce some applications of reconfigurable metasurfaces classified by the approaches to implement their reconfigurabilities.

## 3. Material Technology

Materials that can change states under different external excitations provide new methods for metasurface designs to realize reconfigurable functions.

### 3.1. Liquid Crystal

Liquid crystal, which consists of orderly oriented molecules, is a special material between solid crystal and liquid. When the liquid crystal is excited and affected by electric fields, the liquid crystal molecules will be arranged to the same orientation neatly. When there is no electric field or the liquid crystal is heated, the arrangement will be chaotic and disorderly. Liquid crystal is easy to be integrated with metasurfaces, providing new ways to design optical systems [112,113,114,115,116,117,118,119,120,121,122,123,124,125,126,127]. As shown in Figure 1a [114], Komar et al. designed a transmissive nano silicon reconfigurable metasurface with a liquid crystal layer covering it. At low temperature, the liquid crystals are oriented in parallel; when the temperature increases, the liquid crystal is isotropic and disorderly, and the refractive index of the liquid crystal layer changes. At high temperature, the phase gradient configured by the nano silicon disk realizes the steering of the transmitted beam. Figure 1b [128] introduces an electrically adjustable liquid crystal metasurface. The state of liquid crystal can be changed by applying voltage to the electrode to generate an electric field. Zhu et al. designed nano-scale electrodes to precisely manipulate local electric fields and realized holography in the optical frequency band. When the metasurface is configured with the voltage corresponding to the data information, the transmitted wave will carry this information when passing through the metasurface, and the encrypted optical communication is realized. Compared with commercial spatial light modulators, the metasurface-based optical communication system demonstrated has smaller volume and higher resolution. In Figure 1c, Sharma et al. proposed a metasurface based on liquid crystal chromatic plasmonic polarizers to realize dynamic color manipulations [129]. The plasmonic metasurface consists of rod-shaped and cross-shaped nanoantenna arrays, fabricated on indium tin oxide coated glass substrate. The liquid crystal layer covers the plasmonic metasurface with two glass substrates working as its electrodes. The liquid crystal layer converts incident y-polarized light into x-polarized light at the initial state, and the incident y-polarized light will pass through the liquid crystal layer without any polarization conversion when saturation voltage is applied on the electrodes. The localized surface plasmon resonance (LSPR) on the metasurface is sensitive to the polarization of light. By changing the voltage as well as the polarization of light, the corresponding LSPR wavelength changes, and the color of the nanoantenna arrays is manipulated dynamically. As demonstrated in Figure 1d, Liu et al. designed a liquid-crystal-based transmissive digital coding metasurface to realize beam forming [130]. The multilayer metasurface is a metal–insulator–metal resonator, the two layers of metal work as electrodes, and liquid crystal fills the area between the electrodes. When voltage is applied on the electrodes, the orientation of the liquid crystal rotates. Therefore, the permittivity of the liquid crystal layer changes, resulting in phase differences at different voltage states. The researchers encoded the two states of different bias voltage as binary codes, then calculated the needed code sequences for beam steering and OAM beam generating. This work expands beam forming to the terahertz band and shows its potential applications in terahertz wireless communications.

### 3.2. Phase-change Material

Phase-changing materials have a variety of mechanisms to realize reconfiguration. For example, Vanadium dioxide (VO2) can be transformed from insulator to conductor by increasing the temperature; GST materials, which are composed of germanium (Ge), antimony (Sb), and tellurium (Te), can be heated and cooled to transfer between disordered amorphous states and the ordered crystal state. VO2 [106,131,132,133,134,135,136,137,138,139,140,141,142,143,144,145,146,147,148,149] and GST [105,150,151,152,153,154,155,156,157,158,159,160,161,162,163] provides abundant reconfiguration mechanisms for reconfigurable metasurfaces.

#### 3.2.1. Vanadium Dioxide

Figure 2a [164] shows a reconfigurable metasurface loaded with VO2 material in the terahertz band. At low temperature, VO2 is an insulator, VO2 together with the metal in the upper and lower layers is equivalent to a metal–insulator–metal resonant cavity, the incident wave will interact with the resonant cavity and be absorbed; the absorption rate reaches 90% in a wide band. Similarly, Wang et al. demonstrated a VO2-based metasurface absorber. Figure 2b [165] shows the model and equivalent circuit of the proposed absorber. The VO2 chips are integrated on the surface structure and work as a changeable resistance in the equivalent resonant circuit. When VO2 is a conductor, the chips work at the low resistance state, the circuit resonates at the LC series resonance frequency; when VO2 is an insulator, the LC resonant loop is removed from the whole model, the resonance will move to a higher frequency. Therefore, the abortion at the resonance frequency can be controlled by changing the state of the VO2 chips. Figure 2c [166] shows a novel design for a dual-channel storage device. Lu et al. designed a dual-band reconfigurable metasurface based on VO2. The conductivity of VO2 can be changed when the phase transformation degree of VO2 is controlled under external current stimulations. The transmittance of the metasurface varies independently at two resonance frequencies during the VO2 phase transition process, the low and high transmittance are coded as two storage states of “0” and “1”. Due to the phase transformation hysteresis of VO2, as is shown in Figure 2d, only when the stimulating current changes rapidly will the transformation state change. The “0” and “1” states transformation can be read and erased by encoding current pulses. The combination of two independent channels in two working frequencies realizes a dual-channel memory device.

#### 3.2.2. GST

Figure 3a [167] shows a reconfigurable metasurface loaded with GST material. Two antennas with different shapes are proposed in this work, and the transmission responses of the two antennas are opposite when GST structures are in the same states. Ding et al. designed a metasurface composed of these two antennas to achieve beam deflection in different directions and holograms.

Braid et al. proposed a metalens with a reconfigurable numerical aperture in Figure 3b [168]. This optical metasurface consists of hybrid dielectric-plasmonic GST resonators, microheaters are embedded on the back of resonators to provide tunable thermal phase-change conditions. The metasurface is divided into two parts: an inner circular region to focus light to a point and an outer ring region. In the outer region, resonators are designed to cover 360° phase response and arranged in arrays to provide a spatially flat phase profile. When switching the phase-change states of resonators, the outer region either provides nothing or a spatially flat phase profile, corresponding to high and low numerical apertures.

Zhang et al. realized terahertz wavefront manipulation using a GST embedded metasurface. As is shown in Figure 3c [169], the unit cell is a metal–GST–metal structure where a C-shape split ring resonator is embedded on a GST layer and a metal ground is on the back of the GST layer. Here, the genetic algorithm is applied to optimize the geometric parameters of the unit cell to obtain high polarization conversion efficiency at an amorphous state of GST and low conversion efficiency at the crystalline state. In addition, at the amorphous state, the C-shape resonator is rotated to different angles based on the Pancharatnam−Berry (PB) phase principle and forms a specific phase distribution to realize anomalous deflecting and focusing. The combination of the two designs makes a novel metadevice, which can be switched to anomalous deflecting/focusing with polarization conversion or normal specular reflecting.

Abdollahramezani et al. studied the function of GST with different crystallization fractions in detail [151]. They designed a multi-layer metasurface with tungsten microheaters embedded on the top of the metasurface; the detailed structures are shown in Figure 3d. By applying a customized electrical pulse on the microheaters, a specific crystallization fraction of GST can be realized. The reflection spectra show a pronounced tuning range is achieved upon the multi-state conversion of GST using electrical pulses. This work provides a systematic design to realize chip-scale electrically driven phase-change metadevices.

### 3.3. Graphene

Graphene is a novel two-dimensional material with unique properties that has attracted considerable interest from researchers in the past decades. Graphene has excellent electrical and optical properties. When voltage is applied on the graphene, the Fermi energy levels of graphene will be changed, thereby, changing the conductivity of the graphene. Based on this feature, reconfigurable metasurfaces can be designed [104,170,171,172,173,174,175,176,177,178,179,180,181,182,183,184,185,186,187].

In the terahertz band, how to realize a high reflection with full 360° phase coverage remains a problem. In Figure 4a [188], Wang et al. shows a dynamical reconfigurable metasurface composed of graphene–metal hybrid structures. As the Fermi level increases from 0 to 1.4 eV, the phase differs more than 360° with a reflected amplitude greater than 0.45. Based on this metasurface, beam deflecting and focusing are realized.

Figure 4b [176] shows a metasurface composed of gold nano-antennas loaded with graphene. By configuring the voltages loaded onto the electrodes, the working frequency of the absorption peak (the smallest reflection) can be adjusted. For the two voltage states of −3 V and +7 V tested in this work, the ratio of the reflected amplitude under these two voltage conditions is defined as the modulation depth. When the modulation depth is larger, a larger amplitude change can be obtained by switching the voltage. Based on this principle, a spatial light modulator is designed. When configuring voltages on the metasurface, specific patterns of reflection can be observed, as is shown in Figure 4b.

Similar to reconfigurable metasurfaces based on semiconductor devices, a programmable digitally coded metasurface can also be achieved by loading graphene. Figure 4c [189] shows a digitally coded metasurface. Momeni et al. used this digital coding metasurface to study the relationship between coding states with far field patterns and information entropy and presented a channel encoding strategy for encrypted information transmission.

In Figure 4d [190], Xiao et al. proposed a polarization beam splitter (PBS) based on a graphene metasurface composed of a triple-layer metamaterial made up of two graphene strips and four graphene blocks. When under *y*-polarized illumination, this metasurface performs as an optical switch at six different frequencies in the terahertz band when changing the Fermi levels of graphene on the metasurface. Further research shows the reflection is sensitive to polarization at some of the six frequencies. At 6.143 THz, the metasurface reflects most of the *x*-polarized incident light and transmits most of the *y*-polarized incident light. When linear polarized light incidents, it can be spatially separated into two orthogonal polarizations, thus the PBS is realized.

### 3.4. Liquid Metal

Liquid metal can be configured to form arbitrary shapes and stays at any position in designed structures under precise manipulations. The liquid metal metasurface in Figure 5a [191], with adjustable absorption peaks, achieves nearly 100% absorption in the terahertz band. By applying gas or liquid metal injection, the height of the liquid metal in the cylinder can be adjusted to move the absorption peak. A liquid metal ring metasurface is illustrated in Figure 5b [192]. By adjusting the liquid metal injection and the gas injection, the size, gap direction and gap length of the resonance ring can be adjusted, realizing responses of a 2π reflection phase in broadband. By adjusting the distribution of liquid in the resonant rings, this metasurface enables beam focusing at the same reflection angle in different frequencies and at different angles in the same frequency, which provides a new solution for metasurface, which requires continuous and precise control.

## 4. Electrically Tunable Devices

Since the electromagnetic response of a metasurface can be expressed as an equivalent resistance-inductance-capacitance (RLC) series or parallel circuit, the electromagnetic response can be designed simply by changing the RLC parameters in equivalent circuits. Therefore, the most widely used method to realize reconfigurability is loading semiconductor-integrated devices on metasurfaces and changing the equivalent circuit through electrical control. At present, PIN diodes, varactors and some other devices are widely used to realize a series of reconfigurable metasurfaces.

### 4.1. PIN Diode

PIN diodes are tunable devices. A typical PIN diode works as an on-switch with an inductance and a low resistance when it is forward biased, and when the DC current is zero or reverse-biased, the diode acts as an off-switch with an inductance and a capacitance. When PIN diodes are loaded on the metasurface, two different electromagnetic responses can be achieved by changing the DC bias current, so it is suitable for 1-bit programmable metasurface [193,194,195,196,197,198,199,200,201,202,203]. As shown in Figure 6, there are two typical designs of metasurfaces embedded with PIN diodes, the electromagnetic responses corresponding to Figure 6a [91] and Figure 6b [55] are Figure 6b and Figure 6d, respectively. When the PIN diode is in the “ON” or “OFF” state, the phase difference of 180 degrees between the two states can be reached at working frequency.

The required forward bias current of PIN diodes is generally low and compatible with digital voltage levels, so they can be connected to a field programmable gate array (FPGA) through a bias circuit, realizing simple digital control. Many works were performed to implement programmable digital coding metasurfaces, which directly assign the 1-bit phase distribution on metasurfaces and, thus, change the patterns of the far fields. Figure 7a [62] shows the relationship between the far-field patterns of the metasurface and the coding patterns.

By specifically designing symmetrical metasurface unit cells, two orthogonal PIN diodes can be loaded on the unit cells to respond independently to *x*- or *y*-polarized waves for independent two-dimensional control [69,204,205,206,207]. Figure 7b [208] shows the far-field patterns of a coding metasurface encoded for *x*-polarization and *y*-polarization, respectively. One proper application of manipulating polarizations in two orthogonal directions is polarization conversion. As shown in Figure 7c [204], when dual-linearly polarized waves are incident to the metasurface, only the *x*-polarized waves transmit through the top layer of the metasurface and are converted to *y*-polarized waves in the middle layer. As a result, the transmitted waves are single linear polarized. By changing the state of the PIN diodes, polarization control and conversion can be achieved.

Theoretically, if two PIN diodes are connected in series, four different states can be produced, which can achieve 2-bit digital coding metasurfaces [209,210,211,212,213,214]. Compared to 1-bit metasurfaces, multi-bit metasurfaces have lower quantization error and can achieve more functions due to exhibiting more states. Compared to the 1-bit coding metasurface shown in Figure 7a, the 2-bit coding metasurface shown in Figure 7d [54] is loaded with two PIN diodes in series, performing better in splitting incident waves.

Though the reconfiguration states of PIN diodes are not as abundant as those using varactors, metasurfaces loaded with PIN diodes have their advantages. The loss of PIN diodes is often lower than varactors, resulting in higher efficiency. Furthermore, the design processes are much simpler because fewer parameters are considered in designs. Thus, they still have a wide range of applications. Figure 8a [32] shows a metasurface loaded with PIN diodes that makes efforts toward multi-frequency beam shaping. Castaldi et al. provided a new method to generate various scattering patterns at several selected harmonic frequencies by designing temporally intertwined coding sequences. As illustrated in the figure, researchers demonstrated beam steering, beam scattering and OAM beams generating at four different harmonic frequencies, proving the wavefront manipulation capability of their metasurface. Figure 8b [215] is a typical application in wireless communication systems. Wan et al. proposed a metasurface-based system to integrate user tracking with wireless digital transmission. The experiment shows the metasurface is able to scan, lock the user and send messages to the user simultaneously. Figure 8c [216] illustrates an application of temporal amplitude modulation. Similar to space–time modulation with a time-varying reflection phase, changing the amplitude of reflected waves temporally can also manipulate the frequency spectrum of reflected waves. Wang et al. used PIN diodes to change the active impedance of the metasurface and then change the abortion of incident waves. This platform provided several fake locations in the test radar systems, showing its capability to precisely control each harmonic with suitably designed reflector-absorber switching gaps. Figure 8d [217] shows a novel metasurface performing complex amplitude modulation. With a group of PIN diodes controlling the reflected phase and another group controlling the amplitude, the simultaneous modulation of phase and amplitude is realized. Compared to traditional phase modulated metasurfaces, this proposed metasurface can manipulate the intensities of shaped beams. In the experiment, Liao et al. successfully realized multi-beam shaping by designing the coding patterns on the metasurface.

Zhang et al. further researched space–time modulation [43]. They managed to design 2-bit unit cells with 0-, 90-, 180- and 270-degree phase responses. When setting a specific time-varying coding sequence on the metasurface, any phase response can be obtained as an average of the total coding sequence in time domain. This average phase method provides a new way to achieve multi-bit metasurfaces. As shown in Figure 9a, since any phase vector can be synthesized by a certain two of the basic phase vectors, any phase response in 2π can be realized due to the average in time domain. Figure 9b shows several time-coding sequences and their corresponding phase response. Through specifically designed time sequences, the 2-bit coding metasurface can achieve the equivalent 4-bit phase response. Figure 9c shows a higher bit number of equivalent phase responses can be achieved by increasing the length of coding sequences. Figure 9d illustrates the effect of quantitative errors on metasurface performance, the red line (equivalent 4-bit coding metasurface) achieves a relatively low sidelobe level compared to the 3-, 2- and 1-bit quantized metasurface. The time modulation to achieve an equivalent high-bits metasurface reduces the flexibility on time but greatly reduces the complexity of designing high-bit metasurface unit cells.

Additionally, the PIN diode has some untypical usages on metasurfaces. PIN diodes can be controlled as changeable resistance under different forward bias voltages. Some works are based on this feature to realize dynamic amplitude modulation [109,217,218,219].

### 4.2. Varactor Diode

Varactors are another type of microwave devices, and similarly, they can be equivalent to RLC circuits. When they are reverse-biased, they can be regarded as a capacitor; by increasing the amplitude of the reverse bias voltage applied on the varactor, the capacitance of the varactor decreases gradually. Although continuous capacitance change can be achieved by applying continuous different bias voltage, in practice, it is difficult to obtain multi-channel continuous voltage sources, thus, selecting some capacitance values to design a multi-bit programmable coding metasurface is more practical. However, compared to PIN diodes, the bias voltage required for the reverse bias of varactors is usually higher than the commonly used 5 V or 3.3 V levels in digital circuits [28,30,220], so controlling varactors generally requires extra voltage-controlling circuits or programmable voltage sources. Furthermore, varactors have higher losses than PIN diodes, for example, at approximately 5 GHz, the equivalent resistance of the PIN diodes from [221] is 0.5 Ohm and the equivalent resistance of the varactors from [220] is 2 Ohm. Higher loss from varactors brings some inconvenience to practical applications.

Metasurface loaded with varactors enable more states and more complicated functions [107,222,223,224,225,226,227,228,229,230,231,232,233,234,235,236,237,238,239,240]. In Figure 10a [241], a Huygens metalens is shown. By configuring the voltage of varactors on each row, the incident wave can be focused on any point of the two-dimensional plane in real time. Furthermore, it is possible to split and focus the beam on any two points in the plane. Figure 10b [242] shows a multi-layer metasurface consisting of two layers of quarter-wave plates as the top and bottom layers and a reconfigurable double refractive structure loaded with varactors as the inner two layers. The two-layer reconfigurable structure is orthogonal to each other; the first layer is an adjustable phase shifter for *y*-polarized waves, and it will add a fixed phase shift to *x*-polarized waves, and the second layer is similar; it is adjustable for *x*-polarization and fixed for *y*-polarization. By changing the voltage loaded onto the varactors, a large rotation angle of polarization (up to a maximum of approximately 146 degrees) is achieved.

Similar to PIN diodes, varactors can be used in beam shaping applications. Figure 10c shows an example of beam shaping applications [235]. Shi et al. provided a novel design of reconfigurable metasurface with two diodes on each unit cell. By changing the bias voltage on diodes, the transmitted phase response ranges more than 360° when the amplitude remains nearly unchanged. The difference between maximum and minimum of amplitude is less than 2 dB at a working frequency of 5.35 GHz. Dynamic generation of OAM beams of three different orders is realized under different bias voltage patterns. Figure 10d shows a novel design that combines light excitations with varactors [243]. The proposed metasurface is composed of 36 subarrays loaded with varactors. On the back of each subarray are several photodiodes. When lights illuminate the back of the subarrays, the loaded photodiodes convert the various illumination intensities to different voltage levels, and the voltage will be applied to corresponding subarrays. Therefore, researchers can realize remote control of each subarray through configuring the illumination intensities. In Figure 10d, several applications, such as invisible cloak, illusion and OAM beams are posed to prove their design.

Figure 11a [28] shows a space-time coding metasurface loaded with varactors. Time modulation generates nonlinear harmonics in the reflected waves, and by changing the bias voltages of the diodes for different coding sequences, the harmonics of the ± 1 order can act as a symmetrical or asymmetrical phase response. Figure 11b [30] shows a metasurface designed for communication systems. With the change in the bias voltage on the varactors, the reflection coefficient changes. Researchers chose four states to form 2-bit codes and realized QPSK modulation of incident waves through the metasurface. Researchers received the signal stream from the metasurface and decoded the 2-bit digital information. Using a metasurface can simplify the current communication system framework.

### 4.3. Other Semiconductor Devices

The principle of using PIN diodes and varactor diodes to design a reconfigurable metasurface is to change the parameters of equivalent circuits. Some other semiconductor devices can also achieve similar functions [244,245,246].

The amplitude modulation on the metasurface usually attenuates the amplitude of the incident wave. If it is needed to increase the radiation power of the metasurfaces, we need to increase the power of the illumination source, which will cause more energy loss in the amplitude modulation. A reconfigurable metasurface loaded with adjustable RF amplifiers is illustrated in Figure 12a [247]. By controlling the voltage applied to the adjustable amplifiers, the amplitude of the transmitted wave can be adjusted to more than unity. In this work, four states are digitally coded; the receiver used in this experiment can easily distinguish the differences of four amplitudes. This novel modulation can be used in the communication system with amplitude modulation. In addition to power amplification, amplifiers are often applied due to the feature of one-way transmission. Figure 12b [248] shows a nonreciprocal metasurface based on this feature. Each unit cell of the metasurface is integrated with two amplifiers. When the forward amplifiers work and the backward amplifiers are cutoff, the nonreciprocal one-way transmission is realized, and vice versa. When all the amplifiers are working or cutoff, the states of reciprocal transmission or blockage are realized. Since the amplifiers are digitally controlled, it is very convenient to control the power transmission of this system. Based on the nonlinear feature of the amplifiers, Wang et al. have realized the frequency multiplication of microwaves [249], shown in Figure 12c. Each unit cell of the proposed metasurface is composed of two patches as receiving and transmitting antenna and frequency multiplication circuits. The incident microwave is received by the receiving antennas and coupled to the frequency multiplication circuits, then, the microwave with doubled frequency is radiated by transmitting antenna. In their experiment, the incident 5.1 GHz microwave are multiplied to 10.2 GHz with an efficiency of 85%. Amplifiers can also be used in space–time modulations. Figure 12d [40] shows a novel amplitude modulation metasurface. Based on the space–time modulation theory, periodically changing the amplitude of the transmitted wave will generate different harmonics. Compared with traditional amplitude modulation, this novel work is able to enhance the power of modulated microwaves, which means better signal quality in wireless communication systems.

### 4.4. Micro-Electromechanical System (MEMS)

MEMS, also known as a micro-electromechanical system, is a mechanical system in the micro- or nano-scale whose mechanical structures will change under external excitations. MEMSs have similar manufacturing processes to semiconductor devices. MEMS devices have wider applicable bandwidth than semiconductor devices, ranging from microwave band, to the terahertz, near-infrared and even optical bands [250,251,252,253,254,255,256].

Figure 13a [257] shows how MEMSs loaded on a metasurface work to perform reconfigurable functions. When the voltage is applied on the MEMSs, electrostatic force between the electrodes can pull up part of the grating, changing the shape and period of the entire grating and affecting the electromagnetic response of the metasurface. Shimura et al. designed this metasurface to realize dual refraction in visible light band. Figure 13b [98] shows a reconfigurable metasurface loaded with MEMSs achieving different phase responses. When the voltage applied on the MEMSs is different, the angle of the cantilever changes, so different phase responses can be obtained. By controlling the voltage bias of each column, Cong et al. realized dynamic polarization manipulation of the incident wave and real-time dynamic holography in the terahertz band. Figure 13c [83] is another application of MEMSs in optical imaging systems. When adjusting the voltage applied on the MEMS, the focal point of the metalens will change precisely. Compared with the controlling speed of approximately 10 Hertz (10 times per second) for traditional optical imaging systems, the MEMS metalens system can achieve up to the kHz level of controlling speed. Figure 13d [99] shows the research of implementing a logical operation in the terahertz band using a reconfigurable metasurface loaded with MEMSs. The two parts of the open resonance ring in the figure are independently controlled by two bias sources, and the bias voltage controls the ring suspended or falling. When the mechanical states of the two resonant rings are different, the incident wave can pass through the metasurface, and when the states of the two resonant rings are the same, the incident wave is blocked by the metasurface. The relationship between the states of the two resonant rings and the transmission wave constitutes an “exclusive or” operation. According to this principle, information can be encrypted and transmitted by the metasurface, and for the receiving metasurface, an exclusive or operation will decrypt the information again. This process is a successful attempt of binary digital amplitude modulation communication in the terahertz band, which is of great significance to the research into terahertz communication and optical communication.

### 4.5. CMOS Technique

In recent years, with the rapid demand for highly integrated metasurface systems and applications in the terahertz and even the optical frequency band, some techniques that were originally used in integrated circuits are attempted in metasurface design due to the unqualified performance of traditional semiconductor devices in the high frequency band. The techniques of integrated circuits have brought new opportunities for metasurfaces. Integrated circuits are micro or nano structures loaded with transistors, resistors, capacitors, inductors and other components to realize specific functions. It is proper to etch unit cells of metasurfaces on semiconductor substrates and directly integrate metasurfaces with other integrated circuits to realize reconfigurable functions. The complementary metal oxide semiconductor (CMOS) technique is a common approach for manufacturing integrated circuits. It has the advantages of low power consumption, high speed and high reliability. Using the CMOS technique, billions of pairs of complementary field effect transistors (FETs) can be integrated on a silicon wafer to perform complicated functions. If the metal patterns of metasurfaces are etched on the silicon wafer by chemical vapor deposition or electroplating and connected with the field effect transistor manufactured by the CMOS technique, the structures of the metasurfaces can be changed and various functions can be realized by controlling the on–off states of the FETs. Another advantage of the CMOS technique is that even the circuits fabricated by the outdated 90 nm or 65 nm technique is still relatively small compared with the micron size in the terahertz band, so it is convenient to design multi-bit metasurface unit cells with this technique [258,259].

Venkatesh et al. have used a commercial 65 nm CMOS technique to design reconfigurable metasurfaces on silicon chips in the terahertz band [260]. Figure 14a is a photo of the manufactured CMOS metasurfaces. The four metasurfaces are arranged in a square. Each metasurface has 12 rows and columns so the four metasurfaces have 576 units in total. The active devices (including the control circuit) designed by the researchers are well integrated with the metasurfaces and packaged as a chip. As shown in Figure 14b, if eight switches are integrated on a unit cell, it can achieve up to 256 states in theory. However, considering the symmetry of the structure, the structure in this paper can only achieve 84 different states. Figure 14c,d shows the good performance of the integrated metasurfaces in realizing beam focusing, holography and other functions. In addition, the unit cells designed in this paper have a switching speed of up to 5 GHz, which provides a solution for the design of complicated terahertz metasurface systems.

## 5. Mechanical Techniques

Mechanically reconfigurable metasurfaces refer to those using motors or other mechanical equipment to reconstruct the metasurface to change the electromagnetic responses of the unit cells. As shown in Figure 15a [261], it is a reconfigurable metasurface loaded with micro-motors. The rotation of the motors allows the unit cells to have different orientations, resulting in different reflective phase responses due to PB phase theory. The metasurface designed by the researchers can achieve beam focusing of a circular polarization wave in wide bandwidth, and experiments have shown that the different reflection angles can be measured for different configurations of micro-motors, which verifies the feasibility of this design. Figure 15b [262] is another work that used micro-motors to rotate unit cells of the metasurface for reconfigurability. By rotating the unit cells, a near linear reflection phase can be obtained when the incident wave is circular-polarized. The researchers achieved dynamic beam deflecting to a maximum of 60 degrees by configuring the reflected phase in real time.

Materials are also used in mechanically reconfigurable metasurfaces. Figure 15c [263] shows a multifunctional metasurface composed of arrays of nano-structures embedded on a polydimethylsiloxane (PDMS) substrate. When applying strain to the metasurface, the elastic substrate is mechanically stretched but the nano-structures remain unchanged. As a result, the distance between nano-structures increases; in other words, the period of the metasurface changes. Under this circumstance, the metasurface can be regarded as a dynamic diffraction grating with a tunable period. When the period is less than the working wavelength, the metasurface exhibits transmission to reflection as the wavelength increases; when the period is greater than the working wavelength, after passing through the metasurface, the incident light will diffract into three parts: the 0 and ±1 order diffractions. Since the ±1 order diffractions are dominative, the metasurface works as a beam splitter with a dynamic splitting angle relative to the period.

## 6. Comparison among Different Techniques

In general, the various techniques discussed above are sensitive to the external excitations. When incorporating with metasurfaces, those reconfigurable techniques can change the electromagnetic structures of metasurfaces, resulting in different electromagnetic responses, achieving the reconfigurability of metasurfaces. However, limited by the inherent characteristics of those techniques, each of them has their advantages and disadvantages, proper bands and applications, which need to be comprehensively considered according to the costs and requirements of applications.

The advantage of mechanical reconfiguration is its high linearity, high reliability and low cost. Mechanical structures do not directly participate in the electromagnetic responses of metasurfaces, thereby, they avoid bringing extra loss to metasurfaces. However, mechanical systems are often slower than electrical systems due to their bulky volumes. This disadvantage means traditional mechanical systems cannot be used in small-scale (high frequency) metasurfaces. Due to such properties, they are appropriate for large metasurface antennas.

Before reconfigurable techniques were introduced to metasurface designs, semiconductor devices were widely used in designs of microwave circuits and antenna systems for decades [264,265,266,267,268]. The design of metasurfaces is similar to the design of antennas, which makes it easier to design metasurfaces. For different frequency bands and different applications, a wide variety of semiconductor devices can meet most of the design requirements. Most semiconductor devices are electrically controlled; they can be directly controlled by FPGA through controlling circuits, which also provide convenience for real-time and agile controlling. Semiconductor devices also have their disadvantages. First, limited by the development of semiconductor manufacturing process, the performance of semiconductor devices in high-frequency bands is still not satisfactory, which is manifested by the increased loss or changes in properties in high frequency band. Therefore, the research on the reconfigurable metasurfaces loaded with semiconductor devices is mainly focused on the microwave band. With the progress of semiconductor technology, this problem is expected to be solved in the future; Second, the voltage vs. states curves of some semiconductor devices show nonlinear relationships even in their working frequency band, which brings some inconvenience to the design and control of metasurfaces. The common solution is to convert nonlinear relationships to linear relationships; Third, the working characteristics of semiconductor devices are related to the temperature and humidity of environment, bringing more difficulties to the design of practical systems. Unfortunately, the systems will not work as expected under extreme working conditions.

Similar to semiconductor devices, MEMS devices are able to be integrated with metasurfaces. They not only can be controlled by electrical excitations but also have thermal and optical controlling methods, which provide greater freedom for the designs and applications of metasurfaces. Moreover, MEMS devices change their micromechanical structures under external excitations to realize the manipulation of electromagnetic waves. Therefore, the working frequency band of MEMS devices is wider than that of semiconductor devices. As a developing technique, the switching speed of MEMS devices is relatively slow due to their mechanical structures. The variety of MEMS devices is not abundant, and the cost is relatively high. However, in the high frequency band where semiconductor devices are difficult to apply, MEMS devices are still feasible choices.

CMOS technique allows researchers to design specific circuits as they wish. Therefore, compared with semiconductor devices, the independent design of transistor circuits will have a greater degree of freedom and can remove the frequency band restrictions of the packaged semiconductor devices. Through designing specialized functions, the response speed of the circuit can become faster. By etching the metal patterns of transistor circuits and unit cells of metasurfaces on silicon wafers, a high integration can be realized. Those integrated metasurfaces can be packaged into chips later, which is helpful to promote the commercialization of metasurface systems. The CMOS technique is generally used on submicron and nano scales. When applied to the terahertz band, it is possible to construct complicated functioning circuits. However, when applied to the metasurfaces in the microwave band, the size of the transistor circuit is too small relative to the unit cells in millimeter size. This is a major factor limiting the application of CMOS technique in the microwave band. With the development of next generation communication technology, the advantages of small volume and complicated functions of integrated circuit technology will benefit the metasurface designs in mm-wave and the terahertz band.

The advantage of material technology applied to reconfigurable metasurfaces is that there are many kinds of materials, providing abundant reconfigurabilities for metasurfaces. However, compared with semiconductor devices and MEMS devices, the difficulty in manufacturing remains a problem to solve, and the manufacturing cost is also expensive. Expertise in material technology and fabricating process is needed to design metasurfaces properly. Reconfigurable materials are often harder to excite. For example, VO2, one kind of phase-change materials, needs a temperature of up to 60 degrees centigrade to change the material state. Mg materials need a chemical reaction to change their states. Not only is the transformation speed slow but also it is difficult to meet these reaction conditions in engineering practice. Moreover, how to carry out local, precise and complicated control is also a challenge. For materials requiring thermal and optical control, it is difficult to limit the heat source and light source to illuminate at a specific local area, so it is not suitable for metasurfaces with complicated states. Therefore, the combination of adjustable materials and other reconfigurable technologies is a solution to these problems.

The comparison is summarized in Table 1.

## 7. The Development in the Future

Since reconfigurable techniques were introduced into metasurfaces, the tendency for metasurfaces is to become more complicated and multifunctional. On the basis of reconfigurable metasurfaces, through encoding the phase response of the metasurfaces, researchers have proposed the concept of digital metasurfaces, thus, building a bridge between the digital world and the physical world. The coding state can be deduced from the given far-field distribution by analyzing the corresponding relationships between the coding states and the far-field distribution of the metasurfaces, which simplifies the design of digital coding metasurfaces. Furthermore, when the modulating signals are time-varying, the reflecting waves can be controlled in the frequency domain. Recently, the metasurface shows a tendency for multidisciplinary intersection, according to some of the latest research results [90,269,270,271,272,273,274,275].

Traditionally, researchers derive equivalent circuits from metasurface structures or analyze EM resonant structures to optimize designs [165,243]. Researchers mainly rely on parameter sweeping to find the best needed amplitude or phase responses and then arrange unit cells with different EM responses as specific amplitude or phase profiles to realize complicated functions. This process is very time-consuming, and human engagement is indispensable. Some recent studies have shown machine learning helps to realize the inverse design of metasurfaces [276,277]. After training the learning model, when the needed EM responses are given, the corresponding metasurface structures can be predicted quickly and accurately, which reduces the consumption on time and simplifies the design process.

Great progress has been made in reconfigurable metasurfaces; however, there are still many potential improvements in future reconfigurable metasurfaces.

The first is how to realize more complicated functions. Complicated functions include simultaneous modulation of different parameters of the electromagnetic wave and simultaneous controlling of different multiple wave beams. Current research on the modulation of electromagnetic waves mainly concentrates on amplitude modulation or phase modulation only. Metasurfaces in the future demand simultaneous modulation of amplitude, phase, polarizations, directions, etc., of the incident wave to meet different complicated situations. For some specific applications, such as communication systems, it is necessary to process the incident waves arriving in the same direction but at different times or arriving in different directions but at the same time, which requires fast and sensitive reaction control mechanisms. With the advancement of technology, metasurfaces will achieve more and more complicated functions, and more complicated modulation mechanisms will be discovered with the appearance of new reconfigurable methods.

The second is to solve the performance problems of reconfigurable metasurfaces. The amplitude of the wave decreases after interaction with metasurfaces, mainly due to dielectric loss or loss from semiconductor devices, which causes increased power loss for the metasurface systems. With the limitations of the working band of semiconductor devices, the current research on reconfigurable metasurfaces is mainly focused on the microwave band, yet in the higher frequency band, the loss of semiconductor devices increases, resulting in a worse response for electromagnetic waves. As mentioned above, how to design qualified reconfigurable metasurfaces in the terahertz band with convenient and economical technique remains a problem. In addition, the working bandwidth of reconfigurable metasurfaces is narrow because the reflected response originates from electromagnetic resonance. Some non-resonant metasurfaces were proposed to broaden the bandwidth [278]. Researchers are making efforts for metasurfaces with high performance.

The third is realizing integration and systematization of reconfigurable metasurfaces. With the increasingly high requirements for the control mechanism of the metasurfaces, the design of special control circuits has also become a part of the metasurface designs that cannot be ignored. The integration of the control circuits with metasurface unit cells can reduce the volume of the metasurface. In the past few decades, the development of integrated circuit technology has made the semiconductor device and chip industry prosperous. At present, it is not necessary to understand the internal working principle of the devices, and the application can be carried out directly according to the function of the devices. If a functional reconfigurable metasurface design can be packaged into a chip, it will also help the metasurface to be applied to more fields. In recent years, researchers have made some achievements in the systematization of reconfigurable metasurfaces, designing a metasurface-based communication system that can send and receive data, which is expected to change the framework of the next generation of wireless communication systems. By introducing techniques in related fields, it is hopeful to achieve an adaptive and intelligent systems with metasurfaces.

Finally, the reliability of metasurface is an important issue for practical applications. In the current testing procedure, metasurfaces are often characterized in laboratories and EM chambers once metasurfaces are fabricated. The environments in laboratories are often ideal. In other words, there is a lack of research on aging and decay of metasurfaces. However, the aging of semiconductor devices and material oxidation exists. The influence of metasurfaces aging needs to be evaluated. In addition, metasurfaces are often exposed to ultraviolet light, wind, high temperature and rain in practice. The performance of metasurfaces under tough conditions may be unqualified. For example, the effective permittivity of the substrate of a metasurface changes when it is covered by water. Metasurfaces are expected to be widely used in commercial applications in the future, the reliability of metasurfaces is an important aspect for academic research.

With the development of technologies in some other related fields, the relevant shortcomings in reconfigurable technologies will be compensated, and the performance of reconfigurable metasurfaces will be further improved. Through new detailed research of metasurface integration and systematization, metasurface technology will make further progress in the future.

## Figures and Tables

**Figure 1 nanomaterials-13-00534-f001:**
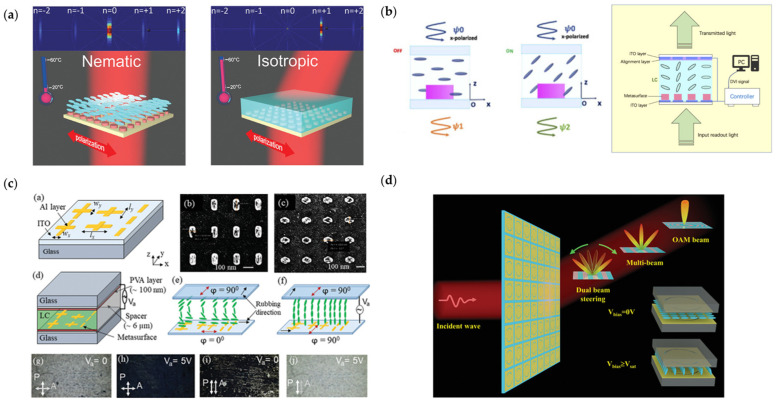
(**a**) Liquid crystal metasurface and its two states while heated. Reprinted with permission from [114], copyright 2023 American Chemical Society; (**b**) Liquid crystal metasurface excited by electric fields, nano-scale electrodes are used to precisely control the configuration states of the metasurface© 2023 Optica Publishing Group. Users may use, reuse, and build upon the article, or use the article for text or data mining, so long as such uses are for non-commercial purposes and appropriate attribution is maintained. All other rights are reserved. (**c**) LC-based plasmonic metasurface with dynamic color controlling. Reprinted with permission from [129], copyright © 2000–2023 by John Wiley and Sons, Inc. or related companies. All rights reserved; (**d**) Terahertz LC-based metasurface realizing beam forming. Reprinted with permission from [130], copyright © 2000–2023 by John Wiley and Sons, Inc. or related companies. All rights reserved.

**Figure 2 nanomaterials-13-00534-f002:**
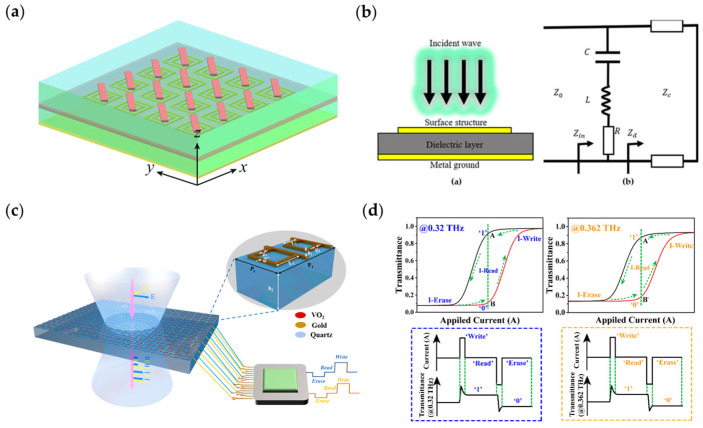
(**a**) Reconfigurable metasurface loaded with VO2 material, the metal–insulator–metal structure forms a resonant cavity. Reprinted with permission from [164], copyright © 2000–2023 by John Wiley and Sons, Inc. or related companies. All rights reserved; (**b**) VO2-based metasurface absorber and its equivalent circuit; (**c**) A novel dual-channel storage device; (**d**) Phase transformation hysteresis of VO2 and the schematic of “write”, “read” and “erase” operation of the storage device.

**Figure 3 nanomaterials-13-00534-f003:**
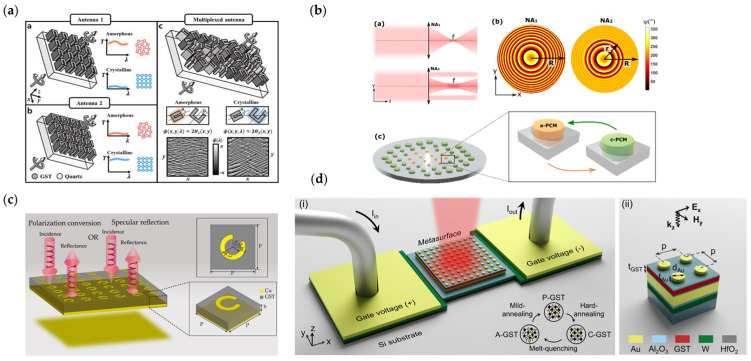
(**a**) Metasurface loaded with GST material realizing beam steering and holography by reconfiguring two complementary nano-antennas. Reprinted with permission from [167], copyright© 2000–2023 by John Wiley and Sons, Inc. or related companies. All rights reserved; (**b**) A metalens with a reconfigurable numerical aperture; (**c**) A novel optical metadevice to realize multifunctional optical responses; (**d**) Reprogrammable metasurface by changing the crystallization fraction of GST.

**Figure 4 nanomaterials-13-00534-f004:**
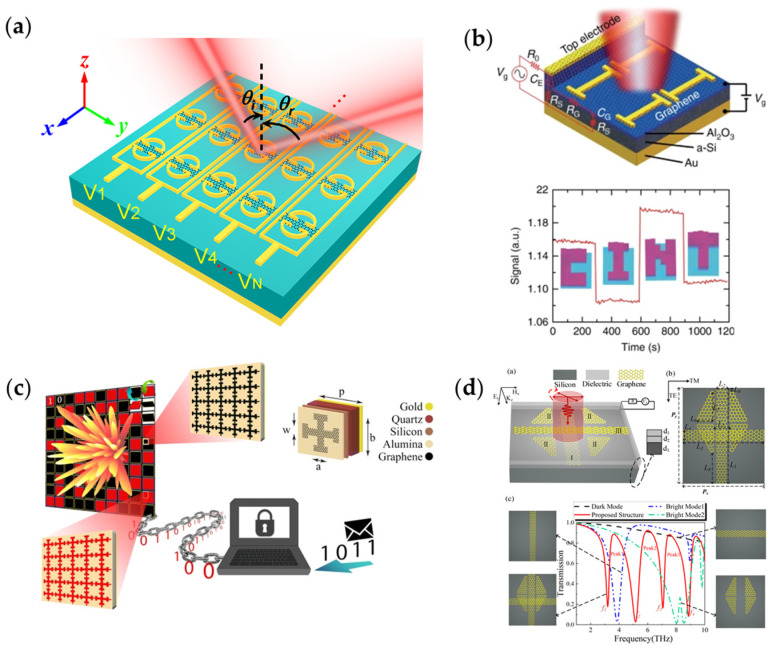
(**a**) Metasurface realizing dynamic 360° phase modulation with relatively high reflectance above 20%; (**b**) Amplitude-modulated graphene metasurface realizing specific near field imaging; (**c**) Graphene coding metasurface and a new encoding strategy based on the metasurface; (**d**) A multifunctional optical device based on graphene to realize optical switching and polarization splitting.

**Figure 5 nanomaterials-13-00534-f005:**
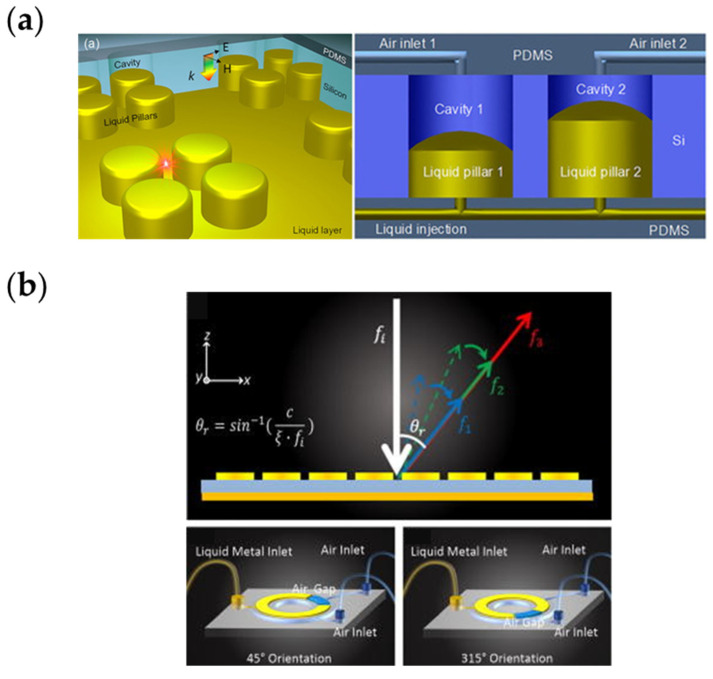
(**a**) Schematic of the metasurface filled with liquid metal; the height of the liquid metal can be controlled precisely. Reprinted from [191], with the permission of AIP Publishing; (**b**) Liquid metal ring metasurface, the geometric parameters can be adjusted by injecting liquid metal or gas. Reprinted from [192], with the permission of AIP Publishing.

**Figure 6 nanomaterials-13-00534-f006:**
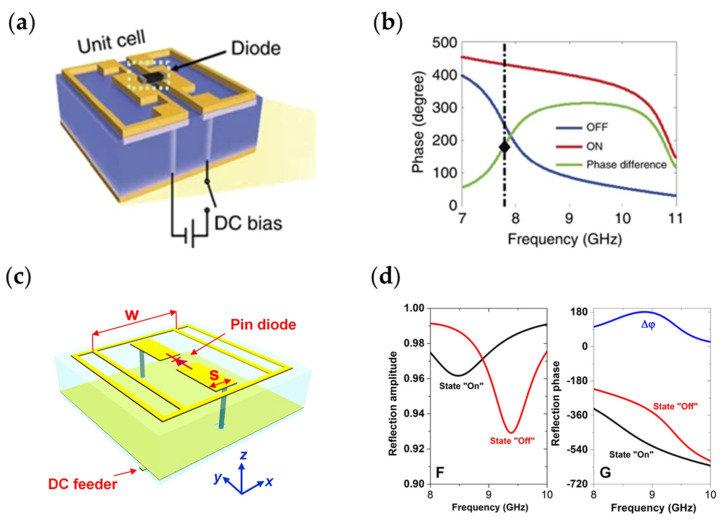
(**a**,**b**) Schematics of two typical metasurface-embedded PIN diodes; (**c**,**d**) The electromagnetic responses corresponding to (**a**) and (**b**), respectively.

**Figure 7 nanomaterials-13-00534-f007:**
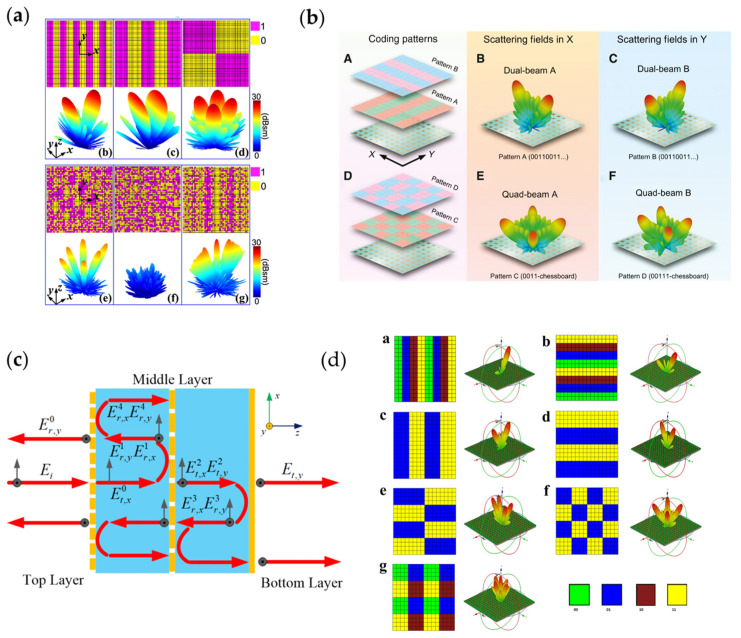
(**a**) 1-bit programmable digital coding metasurface and its far field patterns corresponding to the coding patterns; (**b**) Dual 1-bit programmable digital coding metasurface and its response to x and y-polarized waves; (**c**) Multi-layer 1-bit metasurface to achieve polarization conversion. © 2023 IEEE. Reprinted, with permission, from [204]; (**d**) 2-bit programmable digital coding metasurface and its far field patterns, the far field patterns are more abundant than the 1-bit metasurface.

**Figure 8 nanomaterials-13-00534-f008:**
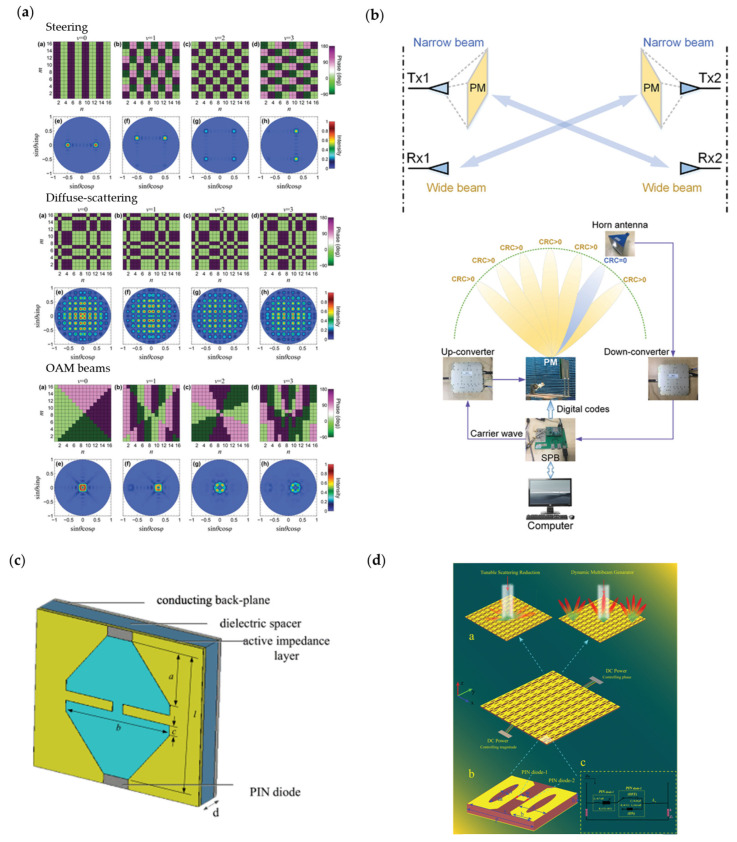
(**a**) Space–time coding metasurface operating at 4 different harmonics. Reprinted with permission from [32], copyright© 2000–2023 by John Wiley and Sons, Inc. or related companies. All rights reserved; (**b**) Metasurface performing user tracking and data transmission. Reprinted with permission from [215], copyright© 2000–2023 by John Wiley and Sons, Inc. or related companies. All rights reserved; (**c**) Amplitude modulation space–time metasurface providing controlled fake targets in radar systems© 2023 IEEE. Reprinted, with permission, from [216]; (**d**) Complex amplitude modulation metasurface that can control both amplitude and reflected angle of beams. Reprinted with permission from [217], copyright© 2000–2023 by John Wiley and Sons, Inc. or related companies. All rights reserved.

**Figure 9 nanomaterials-13-00534-f009:**
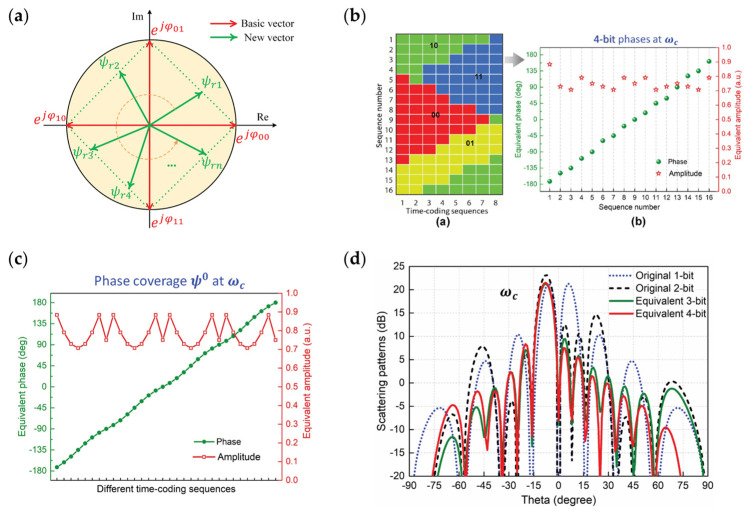
(**a**) Schematic of time modulation on reconfigurable metasurface to achieve any phase response in 2π; (**b**) Equivalent 2-bit phase response achieved by time modulation; (**c**) Realizing of a higher bit number of equivalent phase responses with a longer coding sequence; (**d**) A comparison among the performances of beam steering with equivalent 1, 2, 3 and 4-bit number© 2023 IEEE. Reprinted, with permission, from [43].

**Figure 10 nanomaterials-13-00534-f010:**
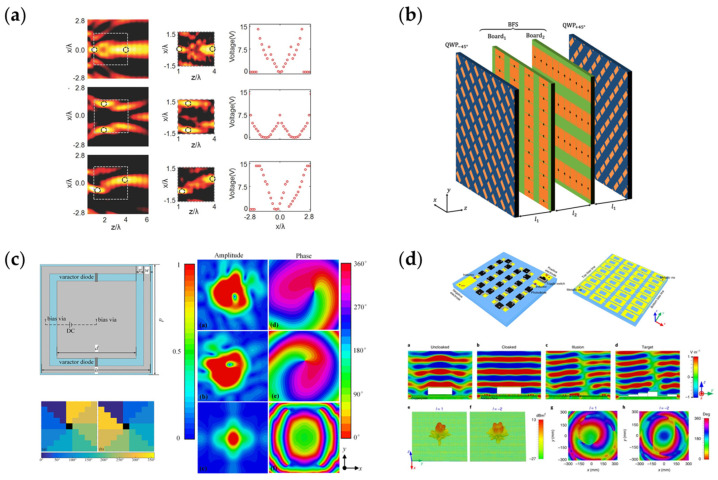
(**a**) A Huygens metalens, the focal point can be selected at any position in two-dimensional plane. Reprinted with permission from [241], copyright© 2000–2023 by John Wiley and Sons, Inc. or related companies. All rights reserved; (**b**) Multi-layer metasurface polarization rotator; (**c**) Metasurface loaded with varactors generating multiple modes vortex beams; (**d**) Light-controlled metasurface which can realize remote control. Reprinted with permission from [243], copyright 2023 Springer Nature.

**Figure 11 nanomaterials-13-00534-f011:**
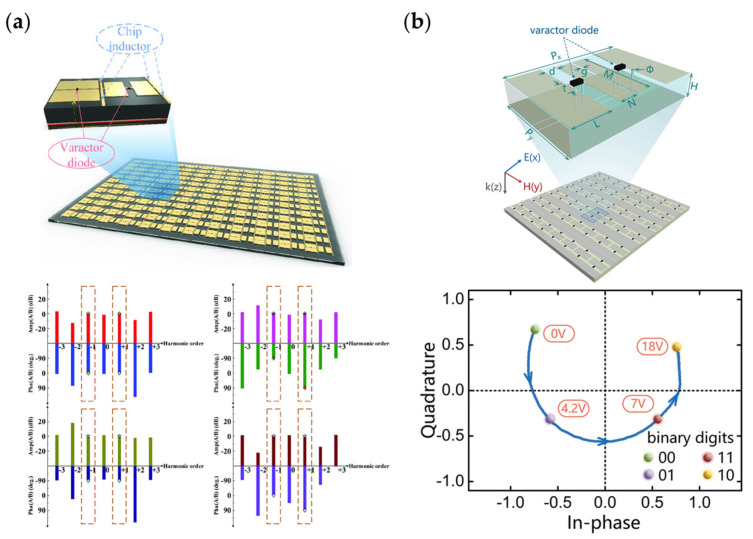
(**a**) Space-time modulation meta to achieve symmetrical or asymmetrical phase response in ±1 order harmonics. Reprinted with permission from [28], copyright © 2000–2023 by John Wiley and Sons, Inc. or related companies. All rights reserved; (**b**) Communication system based on a metasurface to realize QPSK modulation. Reprinted with permission from [30], copyright © 2000–2023 by John Wiley and Sons, Inc. or related companies. All rights reserved.

**Figure 12 nanomaterials-13-00534-f012:**
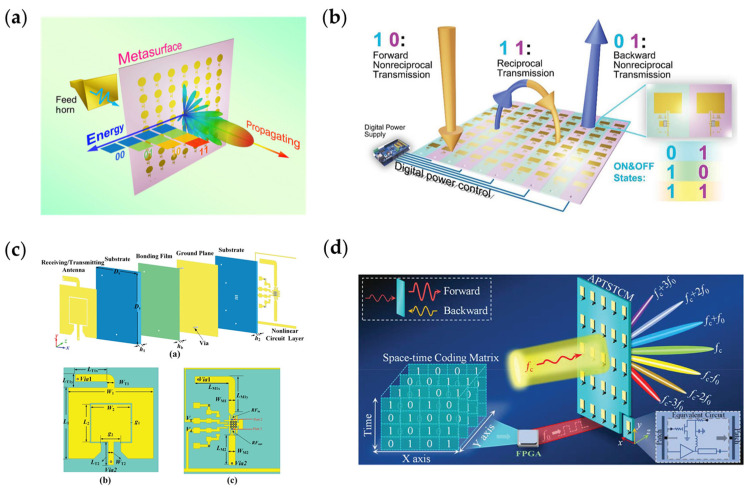
(**a**) Multi-layer metasurface loaded with active RF amplifiers realizing tunable amplitude modulation. Reprinted Figure 1 with permission from [247], copyright 2023 by the American Physical Society; (**b**) Metasurface loaded with amplifiers realizing nonreciprocal transmission. Reprinted with permission from [248], copyright© 2000–2023 by John Wiley and Sons, Inc. or related companies. All rights reserved; (**c**) Metasurface frequency multiplier based on the nonlinearity of amplifiers; (**d**) Space–time metasurface based on amplitude modulation.

**Figure 13 nanomaterials-13-00534-f013:**
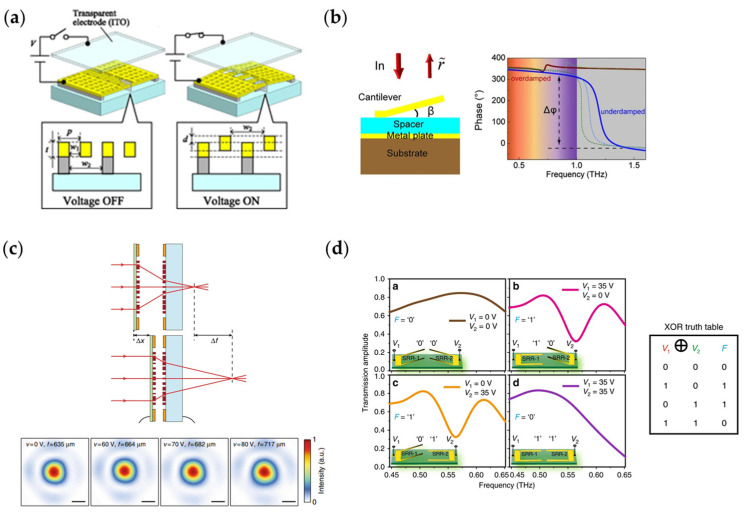
(**a**) Dual refractive metasurface system in the visible light band. Reprinted with permission from [257], copyright 2023 AIP Publishing; (**b**) MEMS reconfigurable metasurface realizing dynamically polarization control and holography. Reprinted with permission from [98], copyright © 2000–2023 by John Wiley and Sons, Inc. or related companies. All rights reserved; (**c**) A metalens imaging system loaded with MEMSs with a changeable focal point; (**d**) MEMS metasurface realizing “exclusive or” operation.

**Figure 14 nanomaterials-13-00534-f014:**
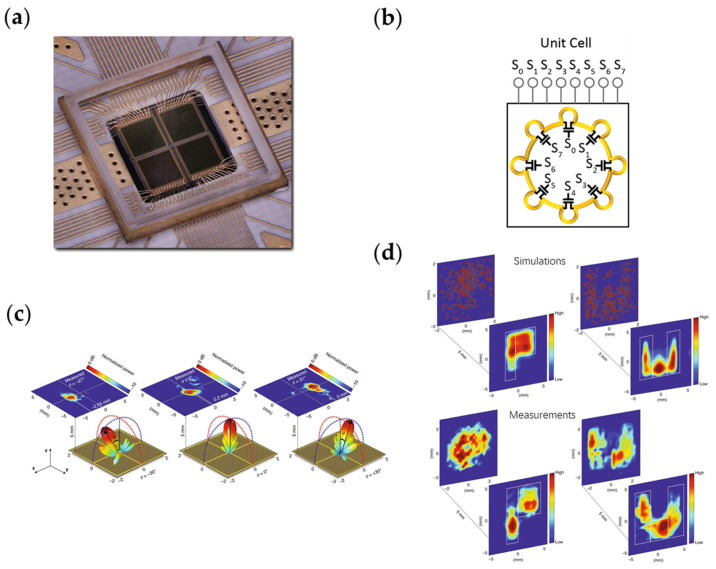
(**a**) A photo of the fabricated CMOS metasurface; (**b**) Schematic of a unit cell from the CMOS metasurface; (**c**) The performance of the metasurface to realize beam focusing; (**d**) The performance of the metasurface to realize holography. Reprinted with permission from [260], copyright 2023 Springer Nature.

**Figure 15 nanomaterials-13-00534-f015:**
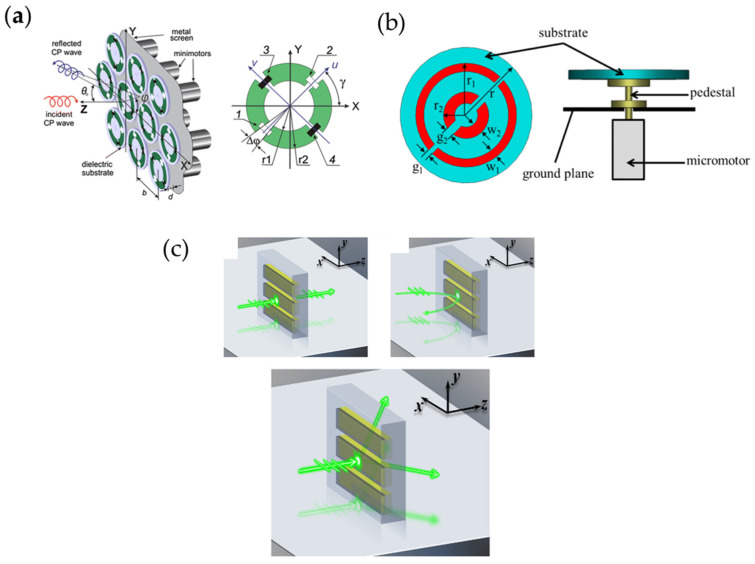
(**a**,**b**) Two proposed works of metasurfaces loaded with micro-motors. Reprinted with permission from [261], copyright 2023 Taylor and Francis Publishing© 2023 IEEE. Reprinted, with permission, from [262]; (**c**) The schematic of the mechanically reconfigurable metasurface.

**Table 1 nanomaterials-13-00534-t001:** Summary on the comparison among different techniques.

Techniques	Frequency	Excitation	Switching Speed *	Cost *	Integration *	Advantages	Defects
Liquid Crystal	THz to optical	Thermal [114]	-	0	+	Wide band	Slow response
		Electrical [128]					
VO2	THz to optical	Thermal [165]	-	-	+	Wide band	Slow response
		Electrical [166]					Precise control
GST	THz to optical	Therma l [168]	-	-	+	Wide band	Slow response Precise control
		Optical [150]					
		Electrical [151]					
Graphene	THz to optical	Electrical [176]	+	-	+	Wide band	/
Liquid metal	GHz to THz	Mechanical [191]	-	0	-	Flexibility	Bulky volume
PIN diode	GHz	Electrical [43]	+	+	+	Simplicity	Loss in high frequency
							Fewer states
Varactors	GHz	Electrical [235]	+	+	+	Simplicity	Loss in high frequency
						More states	
MEMS	GHz to optical	Electrical [99]	0	-	+	Flexibility	Slow response
CMOS	THz	Electrical [260]	+	-	+	Flexibility	/
Mechanical	GHz to optical	Mechanical [263]	0	0	-	High linearity	Bulky volume

* symbols “+”, “0” and “-” refer to good, normal and poor, respectively.

## Data Availability

Not applicable.

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
