# Peer review of "Recent Advances in Reconfigurable Metasurfaces: Principle and Applications"

_nanomaterials, 2023, doi:10.3390/nano13030534_

Round 1

Reviewer 1 Report

The authors present an integrated review on the application and functionality of metasurfaces. A detailed comparison between different techniques such as changeable materials, semiconductor devices, MEMS, CMOS and mechanical equipment is reported. The author also proposed potential future development on metasurfaces. In general, this review presents a clear framework and I recommend it to be published. However, there are large number of sentences lacking proper references.   I suggest the authors to carefully check the grammar before submitting the revised version.

(1) Page 1. From line 23 – 44, there is only 1 citation. For example, line 36, 39 and 44, “… have been reported/realized” but there is no literature to refer to.

(2) Line 289. “Based on” should be in lower case.

(3) Line 334. “there two typical designs” should be “there are two typical designs”.

(4) Line 357. “By change” should be “By changing”.

(5) Line 573, “Recent years” should be “In recent years”.

(6) Line 628, “in Mechanically” should be in lower case.

(7) Line 662, “has its advantages” should be “have their advantages”.

(8) Line 693, change the comma to period in “… band, It …”

(9) Line 753, “Researcher” should be “Researchers”.

(10) Please add reference to the following claims: Line 169-171, 176-178, 202-203, 210-211, 229-231, 376-378, 382-383, 382-383, 387-389, 395-396, 439-442, 549-550, 554-555, 655-657.

(11) There is insufficient discussion on how these surfaces "age" and what is the decay in functionality in various envirnoments.

(12) The description of the liquid crystal surface needs refinement.

(13) There is insufficient discussion on inverse problems, which connects functionality and optmization.

Author Response

We thank the reviewer for the comments on our paper. The detailed responses are in the attachment. Please check the attachment.

Reviewer 2 Report

The paper is well organized and can be considered as a well done review which contains description of different types of metasurfaces and comparison between them. I think it can be published as it is.

Author Response

(The authors gave the same response as above.)

Reviewer 3 Report

The paper widely reviews reconfigurable metasurfaces. The paper is well written and useful for researchers in this community. I have a few minor comments.

1.      As for metasurfaces with graphene, pioneer and important references are missing. Please consider citing the following articles.

[1]    Li, Z.; Yu, N. Modulation of mid-infrared light using graphene-metal plasmonic antennas. Appl. Phys. Lett. 2013, 102, 131108.

[2]    Sherrott, M.C.; Hon, P.W.C.; Fountaine, K.T.; Garcia, J.C.; Ponti, S.M.; Brar, V.W.; Sweatlock, L.A.; Atwater, H.A. Experimental Demonstration of >230° Phase Modulation in Gate-Tunable Graphene–Gold Reconfigurable Mid-Infrared Metasurfaces. Nano Lett. 2017, 17, 3027–3034.

[3]    Ra’di, Y.; Alù, A. Reconfigurable Metagratings. ACS Photonics 2018, 5, 1779–1785.

2.      Many methods are introduced and discussed. If it is possible, please add the table which summarize the benefits and challenges for each method.

Author Response

(The authors gave the same response as above.)

Reviewer 4 Report

Ziyang et al. has written a review of reconfigurable metasurface with special focus on principle and application. Recently, reconfigurable metasurface has been studied widely by different scientific community due to their commercial success and interesting physical properties. Hence, the paper is rightly on time and would be useful for scientific reader interested in this field. However, I feel the paper is written very casually without a systematic approach. There is a plenty of room for improvement and my comments are highlighted below.

1.     There has been recent published review about reconfigurable metasurface by some researcher. I would like to know from the reviewer how is their review is different than the following review. What additional information they cover in this article which has not been covered by other. Some of recent review are mentioned below.

Gu, T., Kim, H.J., Rivero-Baleine, C. et al. Reconfigurable metasurfaces towards commercial success. Nat. Photon. 17, 48–58 (2023)

Saifullah, Y., He, Y., Boag, A., Yang, G.M. and Xu, F., 2022. Recent Progress in Reconfigurable and Intelligent Metasurfaces: A Comprehensive Review of Tuning Mechanisms, Hardware Designs, and Applications. Advanced Science, p.2203747.

He, Q., Sun, S. and Zhou, L., 2019. Tunable/reconfigurable metasurfaces: physics and applications. Research, 2019.

Nemati A, Wang Q, Hong M H, Teng J H. Tunable and reconfigurable metasurfaces and metadevices. Opto-Electron Adv 1, 180009 (2018)

2.     The paper should start with like this, what are the external stimuli’s for reconfigurability and the origin of stimuli’s is it optical, electrical, mechanical, and many more. This is vivid discussion is missing from the paper.

3.     Considering the number of recent review focusing on reconfigurable metasurface, I would expect reviewer to be more focused on aspects the other paper didn’t cover rather than doing the same job. Hence, a figure of merit table is highly necessary to explain the key work and their advantages.

4.     Looking at the application of reconfigurable metasurface, how big is the total market where reconfigurable metasurface finds its application? What are the drawbacks and improvement can be done?

5.     In addition, there are some grammar mistake in the manuscript, and it need to be addressed. 

Author Response

(The authors gave the same response as above.)

Round 2

Reviewer 4 Report

The authors have addressed all of my question and queries properly. Hence, the paper is acceptable at its current form.